physical chemistry

N80 steel, inhibition, PPAPO, chemisorption, endothermic reaction

**Author for correspondence:**
Lei Zhang
e-mail: zhanglei@ustb.edu.cn

# Electrochemical and thermodynamic properties of 1-phenyl-3-(phenylamino)propan-1-one with Na$_2$WO$_4$ on N80 steel

Yun Wang[1], Jun Hu[2], Lei Zhang[1], Jiangli Cao[1] and Minxu Lu[1]

[1]Institute for Advanced Materials and Technology, University of Science and Technology Beijing, Beijing 100083, People's Republic of China
[2]School of Chemical Engineering, Northwest University, 229# North road of Taibai, Xi'an 710069, Shaanxi, People's Republic of China

The corrosion inhibition effect and adsorption behaviour of 1-phenyl-3-(phenylamino)propan-1-one (PPAPO) on N80 steel in hydrochloric acid solution have been investigated by Fourier transform infrared (FTIR), electrochemical method and scanning electron microscopy. The corrosion inhibition mechanism of PPAPO mixed with Na$_2$WO$_4$ was interpreted from the thermodynamic point of view. The results indicated that PPAPO mixed with Na$_2$WO$_4$ acted as a mixed-type inhibitor. The inhibition film formed on N80 steel surface can increase the charge transfer resistance and prevent the occurrence of corrosion reaction, thereby reducing the corrosion rate of metal surface. The inhibition efficiency was up to 96.65%; the inhibitor PPAPO with Na$_2$WO$_4$ showed good synergistic effect on N80 corrosion behaviour in HCl solution. The adsorption behaviour of inhibitors on N80 steel surface was in accordance with the Langmuir adsorption model and mainly belonged to chemisorption. The adsorption process of PPAPO on N80 surface was spontaneous and irreversible endothermic reaction.

## 1. Introduction

The corrosion of carbon steel is a common phenomenon in the oil and gas fields, while it is easy to suffer from the erosion of high active particles in acid environment. However, hydrochloric acid is often used in the solution of acid pickling, chemical cleaning and oil cleaning, removing impurities on the metal surface [1,2]. Chloride ion has the ability to pierce through the surface film due to its small radius and high activity, while the residual chlorine

ion is able to destroy the passive film formed on the surface of metal, developing a large amount of corrosion pits and causing serious corrosion of equipment and piping components. Finally, it may lead to the effect on production and security incidents [3]. Adding inhibitor into corrosion solution is one of the most practical methods to slow down the corrosion rate of metal in some anti-corrosion measures, which has the characteristics of simple process and strong adaptability [4,5].

A few inhibitors added to corrosion medium can slow down the corrosion of metal in acid solution or even worse environment through physical or chemical action [6]. Enormous efforts have been made to research about inhibitors, most of which are toxic and environmentally unfriendly, such as chromate, arsenate and nitrite. Tungstate, which is low toxicity, can be used as a kind of green inorganic inhibitor instead of chromate. Organic inhibitor containing N, S, O, etc., has a certain technical value on the corrosion of metal in acid solution of the process of oil production and chemical processing [7–10]. Han [11] investigated the corrosion inhibition of sodium triphosphate (STPP) on pure aluminium in sodium hydroxide by weight loss method. It was found that the adsorption of STPP on the surface of pure aluminium is the important reason resulted from corrosion inhibition and that the rule of adsorption conforms with Langmuir isotherm. The adsorption process is the endothermic process and the entropy is increased with the rapid increasing of the temperature; the Gibbs free energy steadily decreased, and at the same time, the inhibition efficiency increased. A single inhibitor can prevent the corrosion of metal by reducing the rate of corrosive reaction. However, this may result in lower inhibition efficiency. In recent years, some researchers have extended the research of corrosion inhibitor to compound with other corrosion inhibitors to increase the corrosion resistance. Mu *et al.* [12] investigated the synergistic effect on inhibition by cerium(IV) ion and sodium molybdate for cold-rolled steel (CRS) in hydrochloric acid solution. The results showed that cerium(IV) ion mixed with sodium molybdate produced strong synergistic effect on corrosion inhibition for CRS, and the maximum inhibition efficiency was about 90%. However, the 1-phenyl-3-(phenylamino)propan-1-one (PPAPO) as a base type corrosion inhibitor has the characteristics of stable structure and low toxicity. It has been used in the corrosion of oil pipe casing in acidic medium [13].

In the present work, the inhibition and synergistic effect of PPAPO mixed with $Na_2WO_4$ on N80 steel corrosion were investigated in 0.5 M HCl solution using Fourier transform infrared, potentiodynamic polarization curve, electrochemical impedance spectroscopy and scanning electron microscopy (SEM). Based on corrosion thermodynamics, the corrosion inhibition mechanism was also analysed.

# 2. Material and methods

## 2.1. Materials

N80 steel was used with chemical composition (wt%): 0.42% C, 0.24% Si, 1.55% Mn, 0.012% P, 0.004% S, 0.051% Cr, 0.18% Mo, 0.005% Ni, 0.01% Ti, 0.06% Cu and Fe balance. All specimens were manually mirror-polished by using grit SiC polishing papers with the grit size of 400, 800 and 1000. Then they were washed thoroughly with distilled water, degreased with acetone, washed again with bidistilled water and finally dried at room temperature.

## 2.2. Preparation of sample solutions

The aggressive media was 0.5 M hydrochloric acid solution prepared with 37% hydrochloric acid and distilled water. The PPAPO was synthesized by ketone, aldehyde and amine (schematic diagram is shown in figure 1) [14]. When preparing the PPAPO corrosion inhibitor, 2 mmol of acetophenone, 3 mmol of formaldehyde and 2.6 mmol of aniline were dissolved in 200 ml ethanol solution. The pH value of the solution was adjusted to 3 with HCl aqueous solution. Finally, a red brown liquid was obtained after it was refluxed for 6 h by heating at 60°C. PPAPO and $Na_2WO_4$ were used as corrosion inhibitors. The structure of the synthesized desired product was characterized by Fourier transform infrared (FTIR). The infrared spectrum of the inhibitor was determined by KBr smearing method. The wavenumber of the inhibitor ranged from 4000 to 300 cm$^{-1}$.

## 2.3. Experimental methods

Potentiodynamic polarization and electrochemical impedance spectroscopy (EIS) measurements were conducted in a conventional three-electrode system consisting of N80 steel sample of 0.5 cm$^2$ exposed

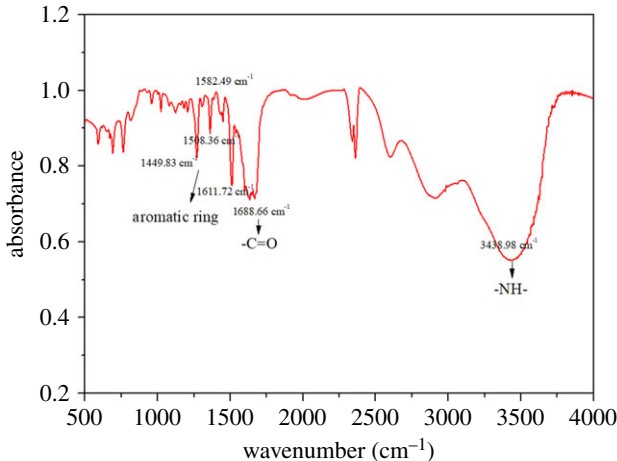

**Figure 1.** Synthetic of PPAPO.

**Figure 2.** IR spectrum of synthesized PPAPO corrosion inhibitor.

area as working electrode, Pt as the auxiliary electrode and a saturated calomel electrode (SCE) as a reference electrode. The experiment was tested at 333 K. Before polarization and EIS measurements, N80 steel electrode was immersed in the corrosive solution until a steady state in open circuit potential (OCP). The experiments were conducted in oxygen-free environment. The scan rate during polarization curve measurements was $0.5 \text{ mV s}^{-1}$ and the potential was scanned from $-250$ to $250 \text{ mV}$ (versus OCP). The EIS was performed at open circuit potential in the frequency range from 100 kHz to 10 mHz with excitation signal of 5 mV.

The morphologies of N80 steel samples in hydrochloric acid in the absence and presence of PPAPO and $Na_2WO_4$ were observed using SEM (VEGA3 TESCAN).

# 3. Results and discussion

## 3.1. Structure characterization of synthetic product

The FTIR spectrum of PPAPO is shown in figure 2.

As shown in figure 2, the strong absorption band centred around $1688.66 \text{ cm}^{-1}$ in the spectrum of PPAPO is attributed to the existence of $-C=O$. The absorption band at $3438.98 \text{ cm}^{-1}$ is assigned to secondary amine N–H stretching vibration absorption peak. Aromatic ring skeleton stretching vibration under normal circumstances has four bands at 1450, 1500, 1585 and $1600 \text{ cm}^{-1}$, which is one of the important marks determining the presence of benzene ring. From figure 2, there are four bands between 1425 and $1650 \text{ cm}^{-1}$, at 1449.83, 1508.36, 1582.49 and $1611.72 \text{ cm}^{-1}$, which are corresponding to the above four band and prove the existence of the benzene ring in the synthetic product. At the same time, a C–H stretching vibration absorption peak of the benzene ring appears at $3095.66 \text{ cm}^{-1}$. Thereby, the results indicate that synthesized red brown liquid product is the PPAPO solution.

## 3.2. Surface morphology

The corrosion morphology of N80 steel specimen in the absence and presence of PPAPO and $Na_2WO_4$ inhibitor are shown in figure 3.

It can be seen that the surface of N80 steel specimen is damaged after immersing in 0.5 M HCl solution in figure 3a. The oxide film on the surface of N80 steel is destroyed by acid solution. On

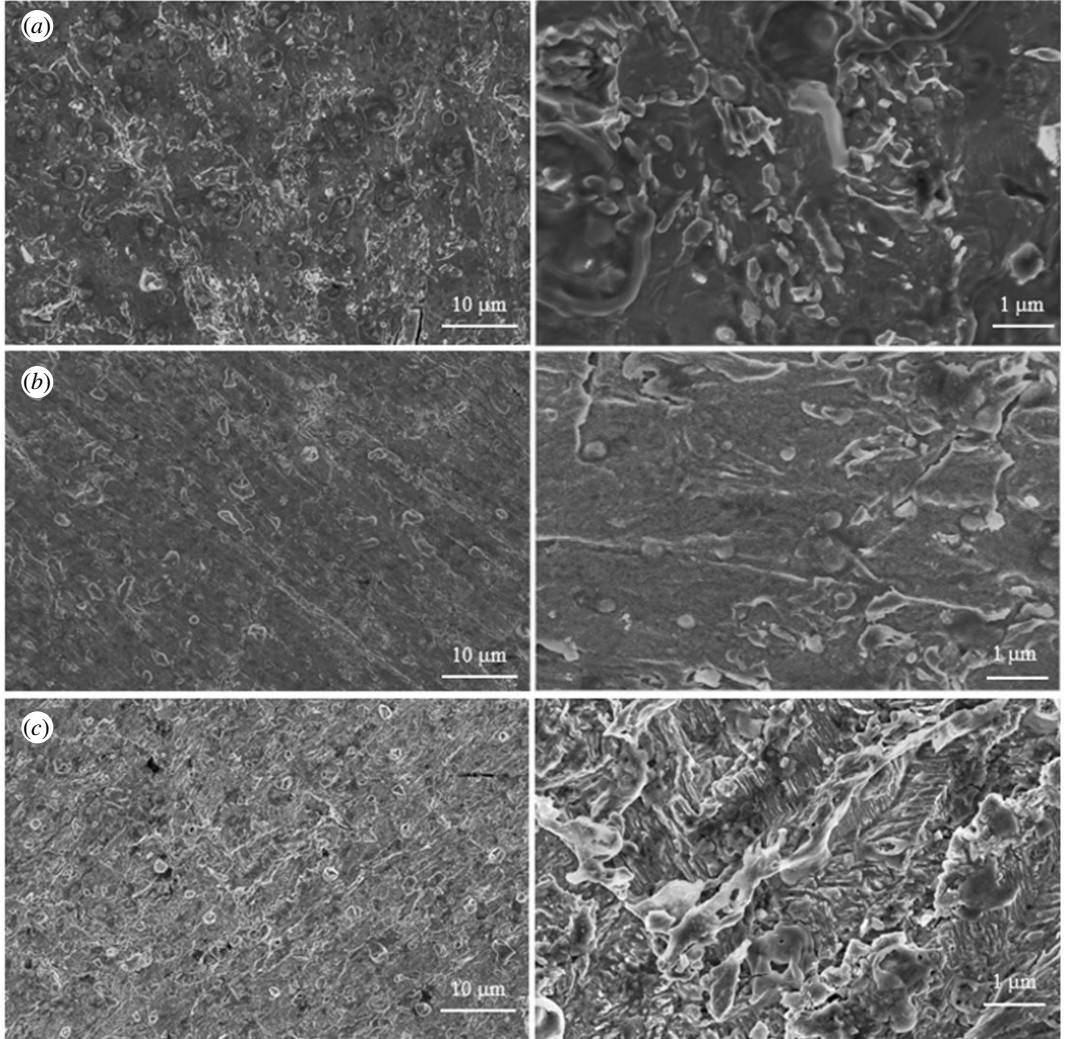

**Figure 3.** Corrosion morphology of N80 in absence and presence inhibitors: (*a*) 0.5 M HCl solution, (*b*) 0.5 M HCl solution + 10 mM PPAPO and (*c*) 0.5 M HCl solution + 10 mM PPAPO + 3 mM Na$_2$WO$_4$.

addition of PPAPO in acidic solution (figure 3*b*), some sediment and minor scratch can be obviously observed on the surface of N80 steel. It indicates that the inhibition film may be formed for PPAPO on the surface of N80 steel, slowing down the corrosion. However, in the presence of PPAPO with Na$_2$WO$_4$ (figure 3*c*), a dense protective layer formed on the N80 steel surface was observed. Meanwhile, the corrosion that occurred on the N80 steel surface was significantly mitigated compared with PPAPO alone.

## 3.3. Potentiodynamic polarization curve

The polarization curves obtained for N80 carbon steel in the absence and presence of varying amounts of PPAPO and Na$_2$WO$_4$ at 333 K are shown in figure 4, and the polarization curve parameters are fitted and listed in table 1. The inhibition efficiency can be calculated through formula (3.1):

$$\eta(\%) = \frac{i_{corr}^0 - i_{corr}}{i_{corr}^0} \times 100,  \tag{3.1}$$

where $i_{corr}^0$ and $i_{corr}$ are self-corrosion electrical current density in the absence and presence of inhibitor, respectively.

As can be seen from figure 4*a*, the polarization curve after the addition of PPAPO corrosion inhibitor in HCl solution moves to the left, which indicates that the corrosion current density has decreased. In addition, the value of $i_{corr}$ gradually reduced with the increase of the PPAPO concentration, which

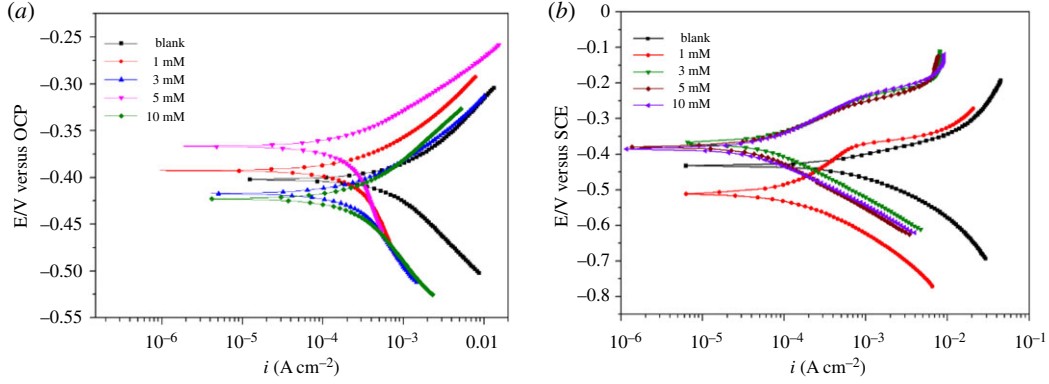

**Figure 4.** Polarization curves of N80 steel in HCl solution containing different concentrations of PPAPO without (*a*) and with (*b*) 3 mM Na$_2$WO$_4$.

**Table 1.** Electrochemical parameters of inhibitors PPAPO with and without Na$_2$WO$_4$ in 0.5 M HCl solution.

| $C_{PPAPO}$ (mM) | Na$_2$WO$_4$ (mM) | $i_{corr} \times 10^4$ (A cm$^{-2}$) | $E_{corr}$ (V versus SCE) | corrosion rate (mm yr$^{-1}$) | $\eta$(%) |
|---|---|---|---|---|---|
| 0 | 0 | 7.76 | −0.41 | 9.08 | — |
| 1 | 0 | 2.83 | −0.39 | 4.81 | 63.53 |
| 3 | | 2.48 | −0.42 | 2.90 | 68.04 |
| 5 | | 2.1 | −0.37 | 2.46 | 72.94 |
| 10 | | 1.95 | −0.42 | 2.28 | 74.87 |
| 1 | 3 | 1.22 | −0.51 | 1.43 | 84.28 |
| 3 | | 0.52 | −0.37 | 0.61 | 93.30 |
| 5 | | 0.42 | −0.38 | 0.49 | 94.59 |
| 10 | | 0.26 | −0.39 | 0.30 | 96.65 |

indicates that the corrosion of carbon steel is inhibited when adding PPAPO. Compared with the results of polarization curve when PPAPO is added alone, the corrosion current density of the mixed PPAPO and Na$_2$WO$_4$ inhibitors (figure 4*b*) is obviously lower than that of PPAPO alone.

As illustrated in table 1, whether sodium tungstate is added or not, the change of corrosion potential $E_{corr}$ is relatively small with the increase of PPAPO concentration. The result indicates that the inhibitor inhibited both the anodic and cathodic processes of corrosion of carbon steel in hydrochloric acid solution. Accordingly, the inhibitor was a mixed corrosion inhibitor. Comparing with $i_{corr}$ of N80 steel in 0.5 M HCl solution without PPAPO, the $i_{corr}$ decreased markedly in the presence of PPAPO. That may because the adsorption of PPAPO on carbon steel surface prevents the anodic metal dissolution and cathodic hydrogen evolution and then decreases the metal surface activation points [15]. The adsorption of inhibitor molecules may include physical and chemical adsorption and generate an energy barrier to block electron transfer. With the increase of PPAPO concentration, the corrosion rate of N80 steel decreases, while the inhibition efficiency of PPAPO increases gradually. When PPAPO was added alone into 0.5 M hydrochloric acid solution, the corrosion rate decreases from 9.08 mm yr$^{-1}$ (0 Mm) to 2.28 mm yr$^{-1}$ as the concentration increases to 10 mM. Meanwhile, the corrosion inhibition efficiency of PPAPO increased to 74.87%. However, the corrosion rate mixed injection of PPAPO and sodium tungstate is lower than that of PPAPO alone, the minimum value of $i_{corr}$ is 0.26 µA cm$^{-2}$ when the PPAPO concentration is 10 mM, and the inhibition efficiency reaches a maximum value of 96.65%. By contrast, it indicates that the addition of Na$_2$WO$_4$ further could inhibit the corrosion of N80 steel in hydrochloric acid.

The Fe atoms on surface of N80 steel may lose two electrons, generating Fe$^{2+}$ when N80 steel corrodes in 0.5 M HCl solution. The H$^+$ in corrosion solution will be reduced to H$_2$ through obtaining electrons. With increasing concentration of PPAPO, the adsorption quantity increases on the surface of N80 steel. Due to the larger adsorption energy of PPAPO molecule than that of tungstate ions [16], the adsorption

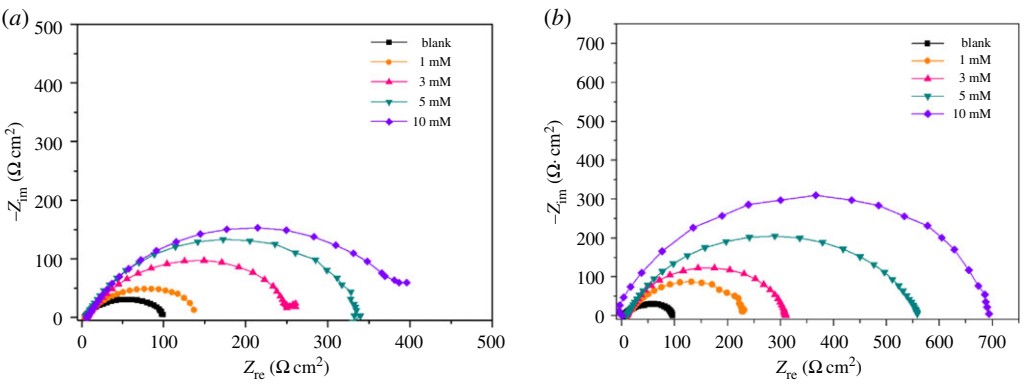

**Figure 5.** Nyquist plots for N80 steel in HCl solution containing different concentrations of PPAPO without (*a*) and with (*b*) 3 mM Na$_2$WO$_4$ at 333 K.

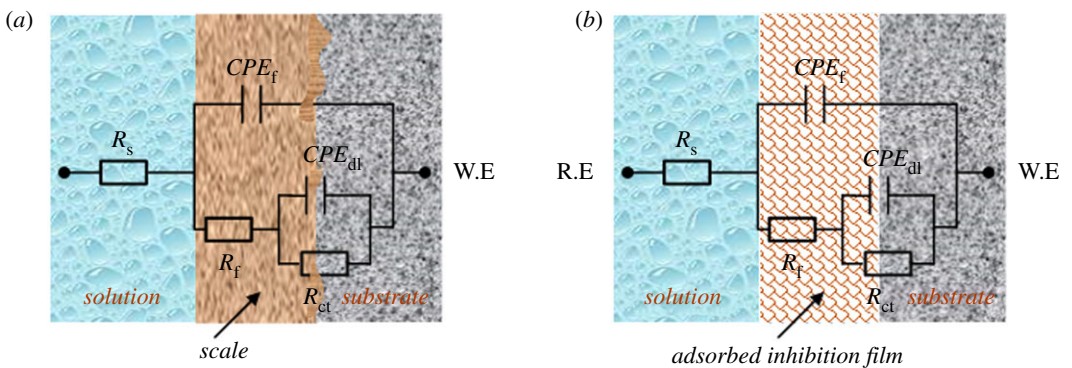

**Figure 6.** Equivalent electrical circuit used to fit the EIS experiment data (*a*) blank, (*b*) PPAPO without and with 3 mM Na$_2$WO$_4$.

centre such as nitrogen and oxygen existing in the structure of PPAPO molecules can preferentially bind to Fe atom on the surface of carbon steel through coordination bond, which covers the surface active area of N80 steel. In addition, WO$_4^{2-}$ ionized by Na$_2$WO$_4$ may precipitate Fe$^{2+}$ near N80 steel electrode through electrostatic attraction (equation (3.2)).

$$WO_4^{2-} + Fe^{2+} \rightarrow FeWO_4 \downarrow . \tag{3.2}$$

In addition, WO$_4^{2-}$ also can react with H$^+$ in corrosion solution to form [WO$_3$(OH)]$^-$, reducing the concentration of hydrogen ions (equation (3.3)):

$$WO_4^{2-} + H^+ \rightarrow [WO_3(OH)]^- . \tag{3.3}$$

Obviously, these results can lead to the decrease of positive charge near anode and change the charge distribution and interfacial properties on the surface of N80 steel. Finally, [WO$_3$(OH)]$^-$ may also compound with Fe$^{2+}$ or Fe$^{3+}$ produced by dissolution of N80 steel and adsorb on metal surface. However, it will increase the energy barrier for electron transfer, slowing down the corrosion of N80 steel in HCl solution. In addition, the poor hydrophilic non-polar groups in PPAPO molecular structure can not only form a layer of hydrophobic membrane due to surface molecular rearrangement, but also hinder the transfer of electrons between the surface of N80 steel and corrosion solution.

## 3.4. Electrochemical impedance spectroscopy

The Nyquist plots for N80 steel obtained at N80 steel and HCl solution interface without and with the different concentrations of PPAPO and Na$_2$WO$_4$ at 333 K are shown in figure 5. The equivalent circuit model as shown in figure 6 is used to fit the EIS data. $R_s$ is the resistance of corrosion solution between the work electrode and reference electrode, $Q_f$ is constant phase element representing the capacitance of corrosion product film, $R_f$ is the resistance of inhibition film, $Q_{dl}$ is constant phase element representing the double-charge layer capacitance and $R_{ct}$ is the charge transfer resistance at

**Table 2.** Electrochemical parameters of N80 steel obtained from EIS Equivalent electrical circuit.

| $C_{PPAPO}$ (mM) | $Na_2WO_4$ (mM) | $R_s$ ($\Omega$ cm$^2$) | $Q_f$ ($\Omega^{-1}$ s$^n$ cm$^{-2}$) | $n_1$ | $R_f$ ($\Omega$ cm$^2$) | $Q_{dl}$ ($\Omega^{-1}$ s$^n$ cm$^{-2}$) | $n_2$ | $R_{ct}$ ($\Omega$ cm$^2$) | $\theta$ |
|---|---|---|---|---|---|---|---|---|---|
| 0 | 0 | 3.36 | 0.00067 | 0.74 | 4.51 | 0.00068 | 0.83 | 90.2 | — |
| 1 | 0 | 3.91 | 0.00044 | 0.92 | 69.75 | 0.00028 | 0.99 | 135.9 | 0.34 |
| 3 | | 10.25 | 0.00045 | 0.83 | 2.793 | 0.00039 | 0.83 | 255.7 | 0.65 |
| 5 | | 10.53 | 0.00041 | 0.83 | 41.11 | 0.00031 | 0.86 | 325.2 | 0.72 |
| 10 | | 6.45 | 0.00008 | 0.88 | 83.91 | 0.00001 | 0.84 | 378.2 | 0.76 |
| 1 | 3 | 9.82 | 0.00050 | 0.82 | 19.08 | 0.00044 | 0.82 | 208.1 | 0.57 |
| 3 | | 9.27 | 0.00007 | 0.89 | 78.23 | 0.00005 | 0.91 | 305.5 | 0.70 |
| 5 | | 9.30 | 0.00003 | 0.93 | 13.17 | 0.00013 | 0.77 | 537.7 | 0.83 |
| 10 | | 10.07 | 0.00002 | 0.97 | 14.14 | 0.00015 | 0.75 | 697.8 | 0.87 |

the metal/solution interface. W.E and R.E represent the working electrode and reference electrode, respectively. The capacitor of the equivalent circuit model must be replaced by a constant phase element (CPE) due to the inhomogeneous metal surface, which consists of the admittance magnitude ($Y_0$) and the exponent ($n$). The impedance of CPE is defined by the expression [17]

$$Y_{CPE} = Y_0(j\omega)^n, \tag{3.4}$$

where $Y_0$ is the magnitude of the CPE; $j$ is the imaginary root; $\omega$ is the angular frequency.

The surface coverage ($\theta$) can be calculated through formula (3.5).

$$\theta = 1 - \frac{R_{ct}}{R_{ct(inh)}}, \tag{3.5}$$

where $R_{ct(inh)}$ and $R_{ct}$ are charge transfer resistance in the absence and presence of inhibitors, respectively.

The result of EIS indicated that the radius of capacitance arc is related to the charge transfer resistance and double electric layer capacitor produced in the process of corrosion [18–20]. As figure 5 shows, there is a capacitive arc in the presence and absence of PPAPO with $Na_2WO_4$ at 333 K, and the change of capacitance arc radius will occur with the concentration of PPAPO. The depressed semicircle in Nyquist plots is related to the non-homogeneity and roughness of the N80 steel surface [21,22]. When N80 steel electrode is immersed in 0.5 M hydrochloric acid, active dissolution of N80 steel will occur. And that is the sample is corroded. With the increase of inhibitor concentration, the radius of capacitive arc increases gradually, which illustrates that the corrosion occurred on N80 steel surface is restrained.

The electrochemical parameters obtained from the experimental data of Nyquist plots are presented in table 2.

The resistance of corrosion reaction mainly comes from the interface polarization between the metal surface and inhibition film [23]. From table 2, the charge transfer resistance increases with addition of different concentrations of PPAPO as compared with the pure HCl solution, namely, the electron transferring between the metal surface and corrosion solution would overcome larger resistance. The increase of $R_{ct}$ value is attributed to the formation of a protective film at the metal/solution interface. However, the value of capacitance for inhibition film and electric double layer decreased after adding inhibitor PPAPO, which may be due to the adsorption of PPAPO at the N80 steel/solution interface. PPAPO molecules can squeeze out the water molecules adsorbed on N80 steel surface, resulting in smaller interface capacitance and larger reaction barrier [24]. With the increase of PPAPO injection concentration, the value of $R_{ct}$ increased from 90.2 to 378.2 $\Omega$ cm$^2$. When PPAPO were added with $Na_2WO_4$ into HCl solution, $R_{ct}$ of PPAPO was higher than that of PPAPO alone at the same concentration and then $R_{ct}$ increased to 697.8 $\Omega$ cm$^2$. Meanwhile, $\theta$ also increases with the increase of PPAPO concentration, which suggests that the inhibition film formed by PPAPO and $Na_2WO_4$ is more uniform. That is also in accordance with the polarization results.

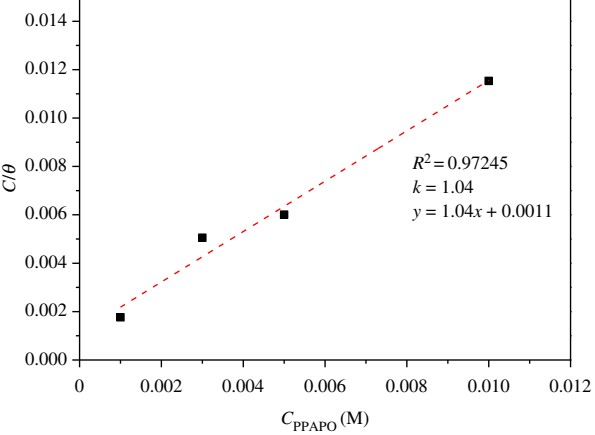

**Figure 7.** $c/\theta \sim c$ plots in different concentration of PPAPO and 3 mM $Na_2WO_4$ at 333 K.

## 3.5. Adsorption isotherm

The surface coverage is an important parameter to study the adsorption behaviour of PPAPO and $Na_2WO_4$ at the metal/solution interface and further analyse the inhibition mechanism. The chemical structure of organic molecules, surface charge distribution and corrosive medium would have an influence on the adsorption process. The plots of $C_{inh}/\theta$ versus $C_{inh}$ for PPAPO in the concentration range 1 to 10 mM at 333 K yielded a straight line with correlation coefficient value 0.9724 (figure 7), suggesting that the adsorption of PPAPO obeys the Langmuir adsorption isotherm represented by the following equation (3.6) [25,26]:

$$\frac{C_{inh}}{\theta} = \frac{1}{K_{ads}} + C_{inh}, \tag{3.6}$$

where $C_{inh}$ is the concentration of PPAPO, M; $K_{ads}$ is the equilibrium constant for adsorption, $l\,mol^{-1}$.

The equilibrium constant is related with the standard free energy of adsorption as follows [27]:

$$K_{ads} = \frac{1}{55.5}\exp\left(-\frac{\Delta G_{ads}}{RT}\right), \tag{3.7}$$

where $R$ is the gas constant; $T$ is the absolute temperature, K; the value of 55.5 is the concentration of water in solution, M; $\Delta G$ is the Gibbs energy change, $kJ\,mol^{-1}$.

The thermodynamic activation parameter, Gibbs energy change ($\Delta G$), was used to investigate the thermodynamic change in the adsorption reaction. The calculated $\Delta G_{ads}$ for PPAPO is $-29.89\,kJ\,mol^{-1}$ at 333 K ($\Delta G_{ads} < 0$), which is between the threshold value for physical adsorption and chemical adsorption, indicating that the adsorbed process for PPAPO on N80 steel surface is spontaneous and also involves both physisorption and chemisorption [28–31].

The heat of adsorption ($\Delta H$) is an important parameter for describing the effect of heat on the adsorption process. It is one of the basic requirements for the characterization and optimization of an adsorption process and is a critical design variable in estimating the performance of an adsorptive separation process [32]. The Arrhenius empirical equation is as follows:

$$K = K_0 \exp\left(-\frac{\Delta H}{RT}\right) \tag{3.8}$$

where $\Delta H$ is the heat of adsorption; $K_0$ is the adsorption equilibrium constant; $R$ is the gas constant ($8.314\,J\,k^{-1}$); $T$ is the absolute temperature (K).

The relationship between surface coverage ($\theta$) and temperature is given according to equations (3.6) and (3.8) [33]

$$\ln\frac{\theta}{1-\theta} = \ln K_0 + \ln c - \frac{\Delta H}{RT}. \tag{3.9}$$

The plot of $\ln(\theta/(1-\theta))$ versus $1/T$ yielded a straight line with correlation coefficient value 0.9741 in figure 8.

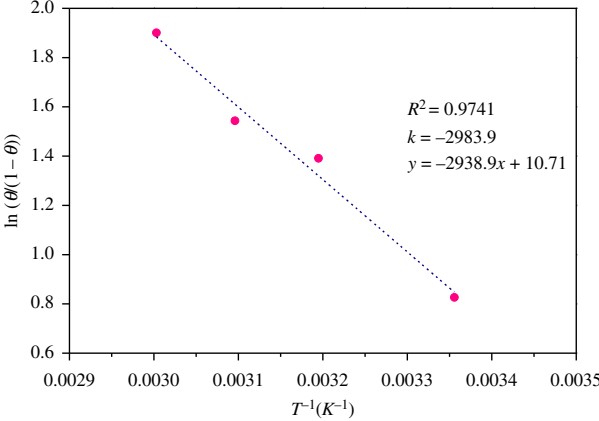

**Figure 8.** ln($\theta/(1 - \theta)$)$\sim$1/T plots in the presence of 10 mM PPAPO and 3 mM Na$_2$WO$_4$.

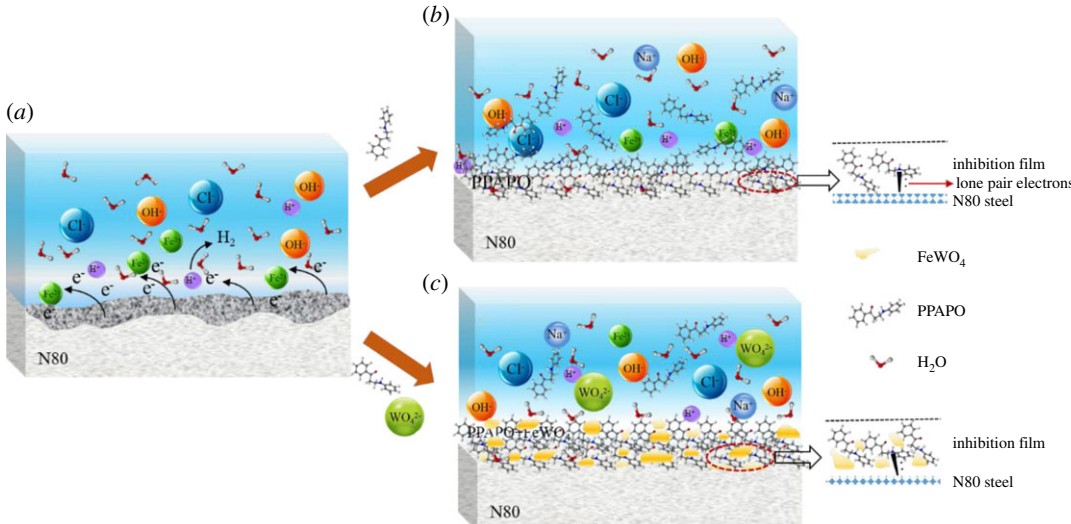

**Figure 9.** Inhibition effect of Na$_2$WO$_4$ and PPAPO on N80 steel in HCl solution: (*a*) HCl solution, (*b*) PPAPO and (*c*) PPAPO and Na$_2$WO$_4$.

The value of $\Delta H$, 24.81 kJ mol$^{-1}$, can be calculated by the slope of ln($\theta/(1 - \theta)$) versus 1/T. It can be concluded that the adsorption behaviour for PPAPO and Na$_2$WO$_4$ on N80 steel surface belongs to endothermic process. At the same time, the value of entropy, 164.2 J mol$^{-1}$ K$^{-1}$, is also obtained through the results of $\Delta H$ and $\Delta G_{ads}$. Finally, the reaction is spontaneous and irreversible.

$$\Delta G_{ads} = \Delta H - T\Delta S. \tag{3.10}$$

# 4. Inhibition mechanism

According to the previous experimental results, the corrosion rate of N80 steel in HCl medium was inhibited by adding PPAPO alone or mixed with Na$_2$WO$_4$, but there is difference for corrosion inhibition mechanism (figure 9).

The corrosion inhibition of Na$_2$WO$_4$ for N80 steel is devoted to the interaction between N80 steel surface and WO$_4$$^{2-}$; a precipitation compound will be formed and deposited on the surface of N80 steel. The O and N atoms with lone pair electrons in PPAPO inhibitor molecules may interact with Fe atom by the coordination bond, neatly arranging on the N80 steel corrosion surface.

When PPAPO was added into corrosion solution alone, the amount of molecules adsorbed on the surface of N80 steel is less, and also the density of inhibition film is smaller. In addition, both WO$_4^{2-}$ and dissolved Fe$^{2+}$ precipitated through electrostatic suction after adding PPAPO and Na$_2$WO$_4$, which is attached to the region uncovered by PPAPO and increases the saturated adsorption capacity of the

metal surface. Furthermore, it prevents the diffusion of electrons from the metal surface to the corrosion solution. Therefore, the compound inhibitors hinder the anodic dissolution and cathodic hydrogen reaction more effectively. At the same time, $Na_2WO_4$ shows a good synergistic effect on the inhibition performance of PPAPO for N80 steel in HCl solution to slow down the corrosion rate; the corrosion inhibition performance increases significantly.

# 5. Conclusion

The corrosion inhibitive properties of PPAPO on N80 steel in 0.5 M HCl solution have been investigated using FTIR, potentiodynamic polarization, EIS measurements and SEM surface analysis.

(i) PPAPO as corrosion inhibitor for N80 steel in HCl solution suppressed the occurrence of anodic metal dissolution and cathodic reaction of hydrogen, thereby reducing the corrosion rate of metal surface. PPAPO is mixed with $Na_2WO_4$ to use for N80 steel corrosion in acid solution, which showed a good synergistic effect. When the PPAPO concentration increases to 10 mM, the highest inhibition efficiency is up to 96.65%.

(ii) The adsorption process belongs to endothermic reaction ($\Delta H > 0$) according to the corrosion thermodynamics analysis of PPAPO and $Na_2WO_4$ with strong coherence to the metal surface. The energy barrier for electron transfer increases with increasing experimental temperature and is conducive to improving inhibitor efficiency. PPAPO mixed with $Na_2WO_4$ significantly reduced the corrosion rate of N80 steel in HCl solution; the process obeys the Langmuir adsorption, and the adsorption is spontaneous process of entropy increase.

Data accessibility. The raw data supporting our paper have been uploaded to the Dryad Digital Repository: https://doi.org/10.5061/dryad.6djh9w0wv [34].

Authors' contributions. L.Z. and J.H. conceived and designed the experiments; Y.W. performed the experiments and analysed the data; L.Z., J.C. and M.L. revised the paper.

Competing interests. We declare that there are no competing interests.

Funding. This work was supported by National Key R&D Program of China (grant no. 2017YFC0805800–02) and Fundamental Research Funds for the Central Universities (grant no. FRF-IC-18-007).

Acknowledgements. The authors thank the postgraduate students for their kind help. The authors also sincerely thank the anonymous reviewers for their constructive suggestions.

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
