## [Reviewer comments · Royal Society Open Science]

Review History

RSOS-191692.R0 (Original submission)

Review form: Reviewer 1

Is the manuscript scientifically sound in its present form?

Yes

Are the interpretations and conclusions justified by the results?

Yes

Is the language acceptable?

Yes

Do you have any ethical concerns with this paper?

No

Have you any concerns about statistical analyses in this paper?

No

Recommendation?

Accept with minor revision (please list in comments)

Comments to the Author(s)

This article chooses 1-phenyl-3-(phenylamino)propan-1-one (PPAPO) as the research object, and explores the corrosion inhibition effect and adsorption behavior of PPAPO on N80 steel in hydrochloric solution. This is a very innovative work, and it is an interesting and quite important for the inhibition behavior of N80 steel in hydrochloric solution. My opinion is the minor revision. However, several issues in the paper should be addressed before the manuscript could be considered for publication.

1. The detailed steps for PPAPO synthesis are unclear.
2. In page 3, for IR spectrum of synthesized PPAPO corrosion inhibitor, some characteristic peaks are discussed in the text, but they are not marked in the figure 2.
3. In the corrosion morphology of N80 in presence inhibitors, why choose the 0.5 M HCl solution containing 2 mM PPAPO?
4. In Equations 2 and 3, the ion subscript needs to be taken seriously, which is inconsistent with the text.
5. In Fig. 6, the meaning of "R.E" and "W.E" needs to be explained in detail.
6. In figure 7 and 8, the equation of fitting curve needs to be added in the figures.
7. There are some grammar issues in the text. They all should be identified and corrected.

Review form: Reviewer 2

Is the manuscript scientifically sound in its present form?

Yes

Are the interpretations and conclusions justified by the results?

Yes

Is the language acceptable?

Yes

Do you have any ethical concerns with this paper?

No

Have you any concerns about statistical analyses in this paper?

No

Recommendation?

Accept with minor revision (please list in comments)

Comments to the Author(s)

The manuscript by Zhang and co-workers reports the anti-corrosion and surface adsorption properties of 1-Phenyl-3-(phenylamino)propan-1-one and Na₂WO₄. The study is well done and contains some significant analytical chemistry. I'm happy to recommend publication subject to the following points being addressed.

I would like to see section 3.3 re-written a little bit. The section starting "As can be seen in Fig. 4..." is unclear. It doesn't appear to be a simple case of 'different concentration of PPAPO' leading to decreases in the corrosion current density. This should be clarified and rewritten.

It may be a question of resolution in the file I received, but Figure 9 is not clear.

On page 8, the English is not clear at all. On the bottom line of the left column 'the organic functional groups in the presence of a pair of electrons...' is confusing and unclear. The reference to the dsp³ hybrid is hard to understand. This could be simplified with a standard explanation of the coordination chemistry.

I would like to see the English significantly improved throughout the manuscript. Sometimes it can be hard to understand the points the authors wish to make.

Decision letter (RSOS-191692.R0)

28-Feb-2020

Dear Professor Lei:

Title: Electrochemical and thermodynamic properties of 1-phenyl-3-(phenylamino)propan-1-one with Na₂WO₄ on N80 Steel
Manuscript ID: RSOS-191692

Thank you for submitting the above manuscript to Royal Society Open Science. On behalf of the Editors and the Royal Society of Chemistry, I am pleased to inform you that your manuscript will be accepted for publication in Royal Society Open Science subject to minor revision in accordance with the referee suggestions. Please find the reviewers' comments at the end of this email. I apologise that this has taken longer than usual.

The reviewers and handling editors have recommended publication, but also suggest some minor revisions to your manuscript. Therefore, I invite you to respond to the comments and revise your manuscript.

Because the schedule for publication is very tight, it is a condition of publication that you submit the revised version of your manuscript before 08-Mar-2020. Please note that the revision deadline will expire at 00.00am on this date. If you do not think you will be able to meet this date please let me know immediately.

- 1) A text file of the manuscript (tex, txt, rtf, docx or doc), references, tables (including captions) and figure captions. Do not upload a PDF as your "Main Document".
- 2) A separate electronic file of each figure (EPS or print-quality PDF preferred (either format should be produced directly from original creation package), or original software format)
- 3) Included a 100 word media summary of your paper when requested at submission. Please ensure you have entered correct contact details (email, institution and telephone) in your user account
- 4) Included the raw data to support the claims made in your paper. You can either include your data as electronic supplementary material or upload to a repository and include the relevant doi within your manuscript

5) All supplementary materials accompanying an accepted article will be treated as in their final form. Note that the Royal Society will neither edit nor typeset supplementary material and it will be hosted as provided. Please ensure that the supplementary material includes the paper details where possible (authors, article title, journal name).

Best wishes,
Dr Laura Smith
Publishing Editor, Journals

On behalf of the Subject Editor Professor Anthony Stace and the Associate Editor Dr Darren Walsh.

RSC Associate Editor:
Comments to the Author:
(There are no comments.)

RSC Subject Editor:
Comments to the Author:
(There are no comments.)

Reviewer comments to Author:
Reviewer: 1

Comments to the Author(s)

This article chooses 1-phenyl-3-(phenylamino) propan-1-one (PPAPO) as the research object, and explores the corrosion inhibition effect and adsorption behavior of PPAPO on N80 steel in hydrochloric solution. This is a very innovative work, and it is an interesting and quite important for the inhibition behavior of N80 steel in hydrochloric solution. My opinion is the minor revision. However, several issues in the paper should be addressed before the manuscript could be considered for publication.

1. The detailed steps for PPAPO synthesis are unclear.
2. In page 3, for IR spectrum of synthesized PPAPO corrosion inhibitor, some characteristic peaks are discussed in the text, but they are not marked in the figure 2.

3. In the corrosion morphology of N80 in presence inhibitors, why choose the 0.5 M HCl solution containing 2 mM PPAPO ?
4. In Equations 2 and 3, the ion subscript needs to be taken seriously, which is inconsistent with the text.
5. In Fig. 6, the meaning of "R.E" and "W.E" needs to be explained in detail.
6. In figure 7 and 8, the equation of fitting curve needs to be added in the figures.
7. There are some grammar issues in the text. They all should be identified and corrected.

Reviewer: 2

Comments to the Author(s)

The manuscript by Zhang and co-workers reports the anti-corrosion and surface adsorption properties of 1-Phenyl-3-(phenylamino)propan-1-one and Na₂WO₄. The study is well done and contains some significant analytical chemistry. I'm happy to recommend publication subject to the following points being addressed.

I would like to see section 3.3 re-written a little bit. The section starting "As can be seen in Fig. 4..." is unclear. It doesn't appear to be a simple case of 'different concentration of PPAPO' leading to decreases in the corrosion current density. This should be clarified and rewritten.

It may be a question of resolution in the file I received, but Figure 9 is not clear.

On page 8, the English is not clear at all. On the bottom line of the left column 'the organic functional groups in the presence of a pair of electrons...' is confusing and unclear. The reference to the dsp³ hybrid is hard to understand. This could be simplified with a standard explanation of the coordination chemistry.

I would like to see the English significantly improved throughout the manuscript. Sometimes it can be hard to understand the points the authors wish to make.

Author's Response to Decision Letter for (RSOS-191692.R0)

See Appendix A.

Decision letter (RSOS-191692.R1)

13-Mar-2020

Dear Professor Lei:

Title: Electrochemical and thermodynamic properties of 1-phenyl-3-(phenylamino)propan-1-one with Na₂WO₄ on N80 Steel
Manuscript ID: RSOS-191692.R1

It is a pleasure to accept your manuscript in its current form for publication in Royal Society Open Science. The chemistry content of Royal Society Open Science is published in collaboration with the Royal Society of Chemistry.

On behalf of the Subject Editor Professor Anthony Stace and the Associate Editor Dr Darren Walsh.

RSC Associate Editor
Comments to the Author:
(There are no comments.)

Reviewer(s)' Comments to Author:

Appendix A

Dear Editor:

Thank you very much for your letter and for the reviewer's comments concerning our manuscript entitled "Electrochemical and thermodynamic properties of 1-phenyl-3-(phenylamino)propan-1-one with Na_2WO_4 on N80 Steel" ID: RSOS-191692. Those comments are very valuable and very helpful for revising and improving our paper, as well as the important guiding significance to our researches. We have studied the comments carefully and tried our best to revise the manuscript. Now we want to resubmit for possible publication in Royal Society Open Science. The main corrections in the manuscript and the responds to the reviewer's comments are as following:

Responds to the reviewer's comments:

Reviewer: 1

Comments:

1. The detailed steps for PPAPPO synthesis are unclear.

Reply: We have re-edited the pictures in the manuscript as follows:

When preparing the PPAPPO corrosion inhibitor, 2mmol of acetophenone, 3mmol of formaldehyde and 2.6mmol of aniline were dissolved in 200 ml ethanol solution. The pH value of the solution was adjusted to 3 with HCl aqueous solution. Finally, a red brown liquid was obtained after refluxed for 6 h by heating at 60 °C.

2. In page 3, for IR spectrum of synthesized PPAPPO corrosion inhibitor, some characteristic peaks are discussed in the text, but they are not marked in the figure 2.

Reply: We have re-edited the Figure 2 in the manuscript as follows:

Fig 2. IR spectrum of synthesized PPAPO corrosion inhibitor.

3. In the corrosion morphology of N80 in presence inhibitors, why choose the 0.5 M HCl solution containing 2 mM PPAPO?

Reply: Thanks for your review, there was a mistake due to carelessness. “(b) 0.5 M HCl solution containing 2 mM PPAPO, (c) 0.5 M HCl solution containing 2 mM PPAPO with Na₂WO₄.” has been modified to “(b) 0.5 M HCl solution + 10 mM PPAPO, (c) 0.5 M HCl solution + 10 mM PPAPO + 3 mM Na₂WO₄.” in manuscript.

4. In Equations 2 and 3, the ion subscript needs to be taken seriously, which is inconsistent with the text.

Reply: we have revised the ion subscript of Equations 2 and 3 in manuscript as follows:

5. In Fig. 6, the meaning of "R.E" and "W.E" needs to be explained in detail.

Reply: The meaning of "R.E" and "W.E" have been added in manuscript as follows: “W.E and R.E represent the working electrode and reference electrode respectively.”

6. In figure 7 and 8, the equation of fitting curve needs to be added in the figures.

Reply: The fitting equation have been added in Figure 7 and 8 as follows:

Fig. 7. $c/\theta \sim c$ plots in different concentration of PPAPo and 3 mM Na_2WO_4 at 333K.

Fig. 8. $\ln(\theta/(1-\theta)) \sim 1/T$ plots in the presence of 10 mM PPAPo and 3 mM Na_2WO_4 .

7. There are some grammar issues in the text. They all should be identified and corrected.

Reply: Thanks for your review, we have checked carefully and modified in manuscript as follows:

“The corrosion inhibition mechanism of PPAPo mixed with Na_2WO_4 was interpreted from the point of view of the thermodynamic. The results indicated that PPAPo mixed with Na_2WO_4 acted as a mixed type inhibitor.”

“The inhibition efficiency was up to 96.65%; and the inhibitor PPAPo with Na_2WO_4 showed good synergistic effect on N80 corrosion behavior in HCl solution.”

“developing a large amount of corrosion pits and causing serious corrosion of equipment and piping components.”

“In the present work, the inhibition and synergistic effect of PPAPPO mixed with Na_2WO_4 on N80 steel corrosion were investigated in 0.5 M HCl solution using Fourier transform infrared, potentiodynamic polarization curve, electrochemical impedance spectroscopy, scanning electron microscopy. Based on corrosion thermodynamics. the corrosion inhibition mechanism was also analyzed.”

“The Fe atoms on surface of N80 steel may lose two electrons, and generating Fe^{2+} when N80 steel corrodes in 0.5 M HCl solution.”

“which consists of the admittance magnitude (Y_0) and the exponent (n).”

“And that is the sample is corroded”

“The surface coverage is an important parameter to study the adsorption behavior of PPAPPO and Na_2WO_4 at the metal/solution interface and further analysis the inhibition mechanism.”

“The equilibrium constant is related with the standard free energy of adsorption as follows”

“where ΔH is the heat of adsorption, K_0 is the adsorption equilibrium constant, R is the gas constant ($8.314 \text{ J}\cdot\text{k}^{-1}$), and T is the absolute temperature (K).”

“The thermodynamic activation parameter, Gibbs energy change (ΔG), was used to investigate the thermodynamic change in the adsorption reaction.”

“According to the previous experimental results, the corrosion rate of N80 steel in HCl medium was inhibited by adding PPAPPO alone or mixed with Na_2WO_4 , but there is difference for corrosion inhibition mechanism (**Fig. 9**).”

“. Further more, it prevents the diffusion of electrons from the metal surface to the corrosion solution.”

“The adsorption process belongs to endothermic reaction ($\Delta H > 0$) according to the corrosion thermodynamics analysis of PPAPPO and Na_2WO_4 with strong coherence to the metal surface.”

Reviewer: 2

Comments:

1. I would like to see section 3.3 re-written a little bit. The section starting “As can be

seen in Fig. 4...” is unclear. It doesn’t appear to be a simple case of ‘different concentration of PPAPO’ leading to decreases in the corrosion current density. This should be clarified and rewritten.

Reply: In section 3.3, we have modified in manuscript as follows:

“As can be seen from **Fig. 4(a)**, the polarization curves after the addition of PPAPO corrosion inhibitor in HCl solution move to the left, which indicates that the corrosion current density has decreased. In addition, the value of i_{corr} gradually reduced with the increase of the PPAPO concentration, which indicates that the corrosion of carbon steel is inhibited when adding PPAPO. Compared with the results of polarization curve when PPAPO is added alone, the corrosion current density of the mixed PPAPO and Na_2WO_4 inhibitors (**Fig. 4(b)**) is obviously lower than that of PPAPO alone.”

“Comparing with i_{corr} of N80 steel in 0.5 M HCl solution without PPAPO, the i_{corr} decreased markedly in the presence of PPAPO. That may due to the adsorption of PPAPO on carbon steel surface prevents the anodic metal dissolution and cathodic hydrogen evolution and then decreases the metal surface activation points [15]. The adsorption of inhibitor molecules may include physical and chemical adsorption and generate an energy barrier to block electron transfer.”

2. It may be a question of resolution in the file I received, but Figure 9 is not clear.

Reply: We have re-edited the picture in the manuscript as follows:

Fig. 9. Inhibition effect of Na_2WO_4 and PPAPO on N80 steel in HCl solution:(a) HCl solution, (b) PPAPO, (c) PPAPO and Na_2WO_4 .

3. On page 8, the English is not clear at all. On the bottom line of the left column ‘ the organic functional groups in the presence of a pair of electrons...’ is confusing and unclear. The reference to the dsp³ hybrid is hard to understand. This could be simplified with a standard explanation of the coordination chemistry.

Reply: Thanks very much for your review, we have modified in manuscript as follows:

“The O and N atoms with lone pair electrons in PPAPPO inhibitor molecules may interact with Fe atom by the coordination bond, neatly arranging on the N80 steel corrosion surface.”

4. I would like to see the English significantly improved throughout the manuscript. Sometimes it can be hard to understand the points the authors wish to make.

Reply: Thanks for your review, we have checked carefully and modified in manuscript as follows:

“The corrosion inhibition mechanism of PPAPPO mixed with Na₂WO₄ was interpreted from the point of view of the thermodynamic. The results indicated that PPAPPO mixed with Na₂WO₄ acted as a mixed type inhibitor.”

“The inhibition efficiency was up to 96.65%; and the inhibitor PPAPPO with Na₂WO₄ showed good synergistic effect on N80 corrosion behavior in HCl solution.”

“developing a large amount of corrosion pits and causing serious corrosion of equipment and piping components.”

“In the present work, the inhibition and synergistic effect of PPAPPO mixed with Na₂WO₄ on N80 steel corrosion were investigated in 0.5 M HCl solution using Fourier transform infrared, potentiodynamic polarization curve, electrochemical impedance spectroscopy, scanning electron microscopy. Based on corrosion thermodynamics. the corrosion inhibition mechanism was also analyzed.”

“The Fe atoms on surface of N80 steel may lose two electrons, and generating Fe²⁺ when N80 steel corrodes in 0.5 M HCl solution.”

“which consists of the admittance magnitude (Y_0) and the exponent (n).”

“And that is the sample is corroded”

“The surface coverage is an important parameter to study the adsorption behavior of PPAPPO and Na₂WO₄ at the metal/solution interface and further analysis the inhibition

mechanism.”

“The equilibrium constant is related with the standard free energy of adsorption as follows”

“where ΔH is the heat of adsorption, K_0 is the adsorption equilibrium constant, R is the gas constant ($8.314 \text{ J}\cdot\text{k}^{-1}$), and T is the absolute temperature (K).”

“The thermodynamic activation parameter, Gibbs energy change (ΔG), was used to investigate the thermodynamic change in the adsorption reaction.”

“According to the previous experimental results, the corrosion rate of N80 steel in HCl medium was inhibited by adding PPAPO alone or mixed with Na_2WO_4 , but there is difference for corrosion inhibition mechanism (**Fig. 9**).”

“Furthermore, it prevents the diffusion of electrons from the metal surface to the corrosion solution.”

“The adsorption process belongs to endothermic reaction ($\Delta H > 0$) according to the corrosion thermodynamics analysis of PPAPO and Na_2WO_4 with strong coherence to the metal surface.”